# BEYOND SOFTMAX: A NATURAL PARAMETERIZATION FOR CATEGORICAL RANDOM VARIABLES

## ABSTRACT

Latent categorical variables are frequently found in deep learning architectures. They can model actions in discrete reinforcement-learning environments, represent categories in latent-variable models, or express relations in graph neural networks. Despite their widespread use, their discrete nature poses significant challenges to gradient-descent learning algorithms. While a substantial body of work has offered improved gradient estimation techniques, we take a complementary approach. Specifically, we: 1) revisit the ubiquitous *softmax* function and demonstrate its limitations from an information-geometric perspective; 2) replace the *softmax* with the *catnat* function, a function composed of a sequence of hierarchical binary splits; we prove that this choice offers significant advantages to gradient descent due to the resulting diagonal Fisher Information Matrix. A rich set of experiments — including graph structure learning, variational autoencoders, and reinforcement learning — empirically show that the proposed function improves the learning efficiency and yields models characterized by consistently higher test performance. *Catnat* is simple to implement and seamlessly integrates into existing codebases. Moreover, it remains compatible with standard training stabilization techniques and, as such, offers a better alternative to the *softmax* function.

## 1 INTRODUCTION

Categorical random variables — random variables that take one of a fixed set of values — are ubiquitous in machine learning. They are used to represent a wide range of concepts, including classes in a classification problem (LeCun & Cortes, 2010), topics in a latent variable model (Miao et al., 2017), discrete actions in a reinforcement learning environment (Mnih et al., 2013), the presence or absence of connections in a graph (Franceschi et al., 2019) and clusters in mixture models (Jacobs et al., 1991).

The use of samples from categorical variables may be a modeling necessity or a choice dictated by scalability and efficiency. For instance, some problems are inherently discrete, such as selecting a word token in a language model (Chen et al., 2018; Paulus et al., 2020) or choosing an action in a reinforcement learning task (Mnih et al., 2013). In other cases, discretization is used for practical reasons, such as scalability in sparse graph modeling (Cini et al., 2023) or information compression using vector quantization in generative models (Van Den Oord et al., 2017).

In many of these settings, the categorical variables are latent, lacking a direct supervisory signal for training. The training signal must therefore be derived from an auxiliary loss function on a downstream task. While low-variance unbiased gradient estimators can be constructed for some continuous latent variables using techniques like the pathwise gradient estimator (Pflug, 1996; Kingma & Welling, 2014), the same methods are often not applicable to the discrete case. Consequently, learning often suffers from high-variance or biased gradient estimates, which can lead to unstable training runs that fail to converge to a satisfactory solution (Peters & Schaal, 2006). As a result, improving the training stability of models with latent categorical random variables remains an active area of research with potential for broad impact (Mohamed et al., 2020; Huijben et al., 2022).

Most techniques developed to stabilize the training of latent categorical distributions focus on reducing the variance of the gradient estimator. This is typically achieved by introducing novel control variates (Gu et al., 2016; Tucker et al., 2017), employing different sampling strategies (Kool et al.,

2020), or designing new gradient estimators (Niepert et al., 2021). In this work we explore a complementary perspective: improving training effectiveness by changing the function that parameterizes the categorical distributions within an information geometry-based framework. The modification we propose is simple to implement, can be easily integrated into existing codebases and is compatible with other training stabilization techniques.

To the best of our knowledge, this is the first work to use results from information geometry (Rao et al., 1945; Amari, 1998) to study the *softmax* parameterization and replace it with a function that has better theoretical properties. Specifically, we observe that the standard *softmax* function has a dense Fisher Information Matrix (FIM), which induces geometric distortions in the parameter space. We therefore propose replacing it with a function designed to produce an optimization landscape more amenable to gradient-descent-based algorithms. We demonstrate that this new parameterization — a series of hierarchical binary decisions that we name *catnat* — yields a diagonal FIM. This diagonal structure substantially reduces geometric distortions, allowing the optimizer to follow a more direct and stable path to a solution. Through extensive experiments in diverse settings − Graph Structure Learning (GSL), Variational Autoencoders (VAEs), and Reinforcement Learning (RL) − we empirically show that the proposed modification enables models to converge to solutions with superior final performance.

## 2 PROBLEM FORMULATION

We consider models that employ a set of latent categorical variables to solve a downstream task. A pipeline general enough to include many deep learning models be described as follows. (a) Given an input $x \in \mathcal{X}$, a neural network $g_\theta$ maps $x$ to a vector of unnormalized scores $\vec{s} \in \mathbb{R}^S$. (b) These scores are transformed by a function $\pi : \mathbb{R}^S \to \Delta^{K-1}$ into a valid categorical probability vector $\vec{p}$ lying in the $(K-1)$-dimensional probability simplex $\Delta^{K-1} := \vec{p} \in \mathbb{R}^K_{\geq 0} : \sum_{k=1}^K p_k = 1$. (c) A latent categorical variable $\vec{C}$ is then sampled according to $\text{Cat}(p_1, \ldots, p_K)$ and (d) used, together with $x$, by a task-specific predictor $f_\psi$ to produce the output $y$:

$$
\begin{aligned}
(a) \quad & \vec{s} = g_\theta(x), \\
(b) \quad & \vec{p} = \pi(\vec{s}) \quad \text{with} \quad \vec{p} = [p_1, \ldots, p_K], \\
(c) \quad & \vec{C} \sim \text{Cat}(p_1, \ldots, p_K), \\
(d) \quad & y = f_\psi(x, \vec{C}),
\end{aligned}
\tag{1}
$$

A training signal is derived from $y$ using a task-dependent objective (e.g., a supervised loss or a reinforcement-learning reward), and the model parameters $(\theta, \psi)$ are learned by gradient-based optimization of the corresponding expected objective. We present the overall architecture in Figure 1, with results naturally extending to more sophisticated architectural variants.

Depending on the application, $g_\theta$, $f_\psi$, or both may be simple neural networks as in VAEs, or compositions of parametric and non-parametric components as in RL. While we focus on a single categorical latent variable for clarity, the formulation and all subsequent results extend directly to collections of categorical variables with potentially different cardinalities.

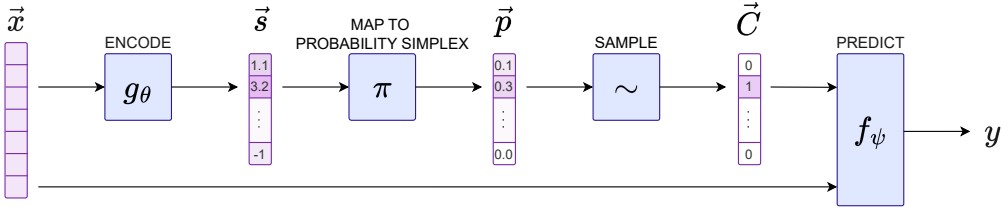

Figure 1: Schematic depiction of a model with a single categorical latent random variable. For rigor, we show $\vec{C}$ as a one-hot vector; however, in some cases (e.g., when using the standard version of the Gumbel–Softmax trick) $\vec{C}$ may be a dense vector.

## 3 RELATED WORKS AND PRELIMINARIES

**Learning Categorical Variables** The most common unbiased gradient estimator is the Score Function, or REINFORCE, gradient estimator (Williams, 1992). While unbiased, it suffers from high variance. This variance can be reduced using control variates (Ross, 2006) by subtracting a baseline from the learning signal. Simple baselines can be constructed by sampling the random variable multiple times, at the cost of introducing non-negligible computational overhead, or they can be estimated as a moving average from previous computations (Kool et al., 2019). More advanced control variates can be built efficiently using a neural network (Mnih & Gregor, 2014; Grathwohl et al., 2018), by employing a Taylor expansion of the mean-field network's loss function (Gu et al., 2016), or by using a low-variance biased estimate of the loss (Tucker et al., 2017). Other gradient estimators can notably reduce variance at the expense of a biased gradient estimate by using a continuous relaxation of one-hot vectors, as in the Gumbel-Softmax (Jang et al., 2017; Maddison et al., 2017; Huijben et al., 2022), or by directly using mean-field gradients as a surrogate (Bengio et al., 2013). In the same class of biased estimators are MAP-based estimators (Niepert et al., 2021; Minervini et al., 2023) that derive a gradient signal from the change in the MAP estimate in response to perturbations of the distribution's parameters.

Instead of changing the gradient estimator another interesting line of research focuses on sampling techniques to reduce the variance of the estimator (Titsias & Lázaro-Gredilla, 2015). For example, Liu et al. (2019) propose to exactly compute the contribution of high probability components and to estimate the rest with an unbiased estimator while Kool et al. (2020) propose to sample without replacement and then unbias the estimate to avoid duplicate samples. Often these techniques can be shown to be a Rao-Blackwellization (Mood et al., 1974) of other more simple estimators.

**Information Geometry & Natural Gradient** Ordinary gradient descent assumes an Euclidean geometry of the parameter space. In Amari (1998) and Amari & Douglas (1998) the authors recognized that the parameter space of many learning models is not Euclidean and equal changes[1] in the parameter space can have disproportional impacts on the model's output distribution. To address this, they proposed measuring the 'distance' between parameter settings through the dissimilarity of their induced distributions, assessed by their Kullback–Leibler divergence. For distributions $p(x|\theta)$ and $p(x|\theta + d\theta)$ close in the parameter space, the KL divergence can be approximated as:

$$D_{KL}(p(x|\theta)||p(x|\theta + d\theta) \simeq \frac{1}{2}d\theta^T G(\theta)d\theta \qquad (2)$$

Where $G(\theta)$ is the Fisher Information Matrix (FIM):

$$G(\theta) = \mathbb{E}_{p(x|\theta)}\left[(\nabla_\theta log\ p(x|\theta))(\nabla_\theta log\ p(x|\theta))^T\right] \qquad (3)$$

The FIM captures the local curvature of the statistical manifold (Amari, 2016) and the natural gradient $\tilde{\nabla}\mathcal{L}(\theta)$ is defined as the direction of steepest descent in this Riemannian manifold. The natural gradient $\tilde{\nabla}\mathcal{L}(\theta)$ can be obtained by pre-conditioning the ordinary gradient with the inverse of the FIM:

$$\tilde{\nabla}\mathcal{L}(\theta) = G(\theta)^{-1}\nabla\mathcal{L}(\theta) \qquad (4)$$

While theoretically advantageous, implementing natural gradient descent presents practical challenges. First, for each update step, computing Equation (4) requires calculating and inverting the FIM, which entails cubic scaling and can easily become a computational bottleneck. For this reason, different approximations have been proposed (Pascanu & Bengio, 2013; Grosse & Martens, 2016; Amari et al., 2019) to speed up computation at the expense of precision. As a second problem, Equations (2) and (4) hold true for infinitesimal parameter changes, so their approximation error increases with larger, more practical step sizes commonly used during optimization. In Martens (2020) natural gradient descent is analyzed from the perspective of a second-order optimization method, demonstrating that designing a robust natural gradient optimizer necessitates the incorporation of techniques such as trust regions and Tikhonov regularization. Furthermore, the FIM can be singular or create numerical instabilities during its inversion. In this work, we propose to tackle both problems by choosing a suitable parameterization for the categorical latent random variable that intrinsically produces a diagonal FIM.

---

[1] measured by some kind of Euclidean norm.

# 4 CATEGORICAL RANDOM VARIABLES PARAMETERIZATIONS

## 4.1 THE SOFTMAX FUNCTION AND ITS PITFALLS

Let $\vec{s} \in \mathbb{R}^{1 \times K}$ be a set of scores, the *softmax* function is defined as:

$$p_i = \frac{e^{s_i}}{\sum_{k=1}^{K} e^{s_k}} \tag{5}$$

Originally introduced in statistical mechanics as the Boltzmann distribution (Jaynes, 1957), the *softmax* later appeared in statistics as the canonical link for categorical outcomes in Generalized Linear Models (GLM) (Nelder & Wedderburn, 1972). In neural networks, the name *softmax* was popularized by Bridle (1989), who also describes some of its appealing properties: it converts arbitrary real vectors into non-negative probabilities, preserves rank order, and offers a smooth approximation to the $\arg\max$. These features have made it the standard parameterization for categorical variables in machine learning.

Despite its benefits, the *softmax* function has non-negligible drawbacks. It is overparameterized, using $K$ parameters to represent a $(K-1)$-dimensional simplex, it can saturate, leading to vanishing gradients (Goodfellow et al., 2016) and, in highly nonlinear probabilistic models, the GLM assumptions behind its usefulness (Nelder & Wedderburn, 1972) may not hold.

We argue that Information Geometry (Amari, 1998; 2016) provides a principled framework for defining more suitable parameterizations. To this end, in Proposition 4.1 we analyze the geometric properties induced by the *softmax* function.

**Proposition 4.1.** *The Fisher Information Matrix for a categorical random variable parametrized by the softmax function, as defined in (5), is*

$$G_{softmax}(s) = \begin{bmatrix} p_1(1-p_1) & -p_1 p_2 & \cdots & -p_1 p_K \\ -p_2 p_1 & p_2(1-p_2) & \cdots & -p_2 p_K \\ \vdots & \vdots & \ddots & \vdots \\ -p_K p_1 & -p_K p_2 & \cdots & p_K(1-p_K) \end{bmatrix}. \tag{6}$$

We provide a proof of the proposition in Appendix A.

The resulting Fisher Information Matrix is dense, with off-diagonal entries $-p_i p_j$ that couple all scores. Consequently, the statistical manifold is curved, and, as discussed in Section 3 and by Amari & Douglas (1998); Amari (2016), gradient-based optimization can suffer. To address this, we introduce a class of parameterizations designed to induce a flatter statistical manifold.

## 4.2 CATNAT: A CLASS OF NATURAL PARAMETERIZATIONS

In this section, we propose a class of parameterizations for categorical random variables that, as demonstrated in Theorem 4.2, yield a diagonal Fisher Information Matrix. We refer to this class as *catnat*, as it parametrizes the categorical distribution in accordance with natural gradient principles. By analyzing the general form of the FIM, we further identify and select the parameterization with the minimal number of factors in the diagonal terms.

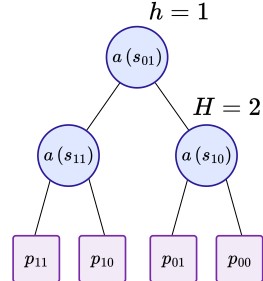

Figure 2: *catnat* parameterization for a categorical distribution with $K = 4$ classes.

The proposed class models the categorical probability distribution as the outcome of a sequence of binary decisions, structured as a hierarchical tree. Each unnormalized score $s_i$ corresponds to a unique node in this tree. To ensure that the resulting probabilities lie in the interval $[0, 1]$, an activation function $a : \mathbb{R} \rightarrow [0, 1]$ is applied to each score. Figure 3 illustrates this construction.

Each score $s_i$, its corresponding binary probability $a_i := a(s_i)$, and the final categorical probabilities $p_k$ are associated with unique indices. To ease the theoretical analysis, we express those indices in their binary representation that, for a categorical distribution over $K$ classes, can be represented by a string of length $H = log_2(K)$ bits. See Figure 2 for an example with $K = 4$.

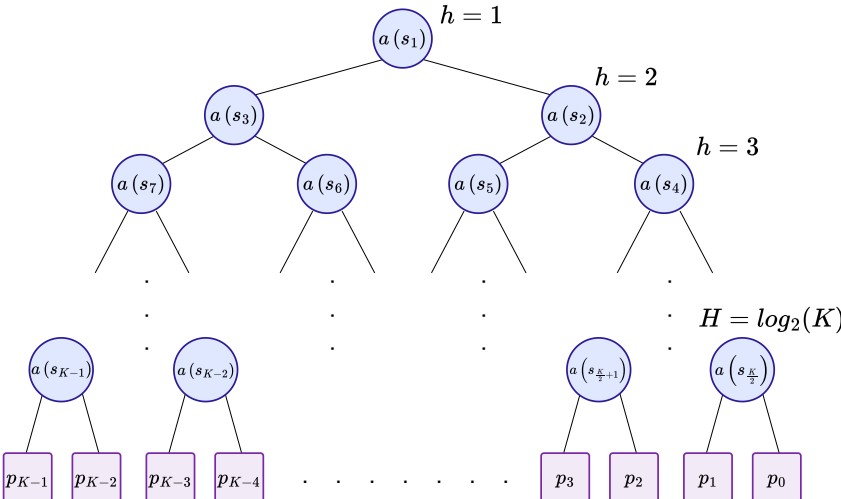

Figure 3: *catnat* parameterization for the categorical distribution. Given unnormalized scores $s_i$ and activation function $a$, blue nodes compute the probability of going left ($a(s_i)$) or right ($1 - a(s_i)$). Final categorical probabilities are shown in purple. On the right side, the hierarchy level $h$ is indicated. Note that indices for $p$ start from zero, while indices for $s$ start from one.

For Bernoulli probabilities $a_i$ and scores $s_i$, we introduce a convenient notation by splitting the binary string into two sequences: $\boxed{\text{HRC}}$ and $\boxed{\text{ID}}$ . Given the hierarchy level $h$ of $a_i$ in the tree, $\boxed{\text{HRC}}$ consists of $h - 1$ zeros followed by a one, while $\boxed{\text{ID}}$ specifies the position of the node at that level. For example, in Figure 4, $\boxed{\text{HRC}}$ shows that $a_9$ is at hierarchy level $h = 3$, and $\boxed{\text{ID}} = 010$ identifies it as the second node from the right.

Since $\boxed{\text{HRC}}$ is uniquely determined once $\boxed{\text{ID}}$ and $K$ are given, we drop $\boxed{\text{HRC}}$ and write, for a node at hierarchy $h$:

$$a_{\boxed{\text{HRC}}\boxed{\text{ID}}} = a_{\boxed{\text{ID}}} = a_{\boxed{b_1, ..., b_{h-1}}}. \tag{7}$$

The introduced notation allows for a compact representation of the categorical probabilities. Specifically, the probability $p_k$ of category $k$ identified by the binary string $\vec{b} = [b_1, ..., b_H]$ is:

$$p_{\vec{b}} = p_{b_1, ..., b_H} = \prod_{h=1}^{H} \left( a_{\boxed{b_1, ..., b_{h-1}}} \right)^{b_h} \left( 1 - a_{\boxed{b_1, ..., b_{h-1}}} \right)^{1-b_h} \tag{8}$$

$$a_9 = a_{\overbrace{001\underbracket{\vphantom{0}010}}}^{H = log_2(K) \text{ bits}}$$
$$\quad\quad\quad \boxed{\text{HRC}} \; \boxed{\text{ID}}$$
$$= a_{\boxed{\text{HRC}}\boxed{\text{ID}}}$$

Figure 4: Example of a binary representation of a node for $K = 64$.

At the root node ($h = 1$), the path represented by the set $b_1, ..., b_{h-1}$ is empty, consistent with the fact that the node is uniquely identified by its hierarchy level alone. By construction, the probability of descending from the root to a node $a_i$ is:

$$P(a_i) = P\left( a_{\boxed{\text{ID}}_i} \right) = P\left( a_{\boxed{b_1, ..., b_{h_i-1}}} \right) = \prod_{h=1}^{h_i-1} \left( a_{\boxed{b_1, ..., b_{h-1}}} \right)^{b_h} \left( 1 - a_{\boxed{b_1, ..., b_{h-1}}} \right)^{1-b_h} \tag{9}$$

This probability is also equivalent to the sum of the probabilities of all leaf nodes $p_{\vec{b}}$ that descend from the node $a_i$:

$$P(a_i) = \sum_{\vec{b} \in \mathcal{D}_i} p_{\vec{b}} \tag{10}$$

where $\mathcal{D}_i$ is the set of all leaf nodes in the subtree rooted at $a_i$.

**Theorem 4.2.** *The Fisher Information Matrix $G_a(s)$ for the catnat parameterization is:*

$$G_a(s)_{ij} = \begin{cases} 0 & if \quad i \neq j \\ P(a_i) \left( \frac{\partial a_i}{\partial s_i} \right)^2 \left( \frac{1}{a_i(1-a_i)} \right) & if \quad i = j \end{cases} \tag{11}$$

The proof is provided in Appendix B. Theorem 4.2 shows that this class of parameterizations yields diagonal FIMs, thereby flattening the statistical manifold. The diagonal entries, $G_a(s)_{ii}$, depend on two components: the probability of reaching node $i$, $P(a_i)$, and a term involving the derivative of the chosen activation function. Since $P(a_i)$ is determined by the scores of all ancestor nodes, each diagonal entry depends on at most $H = \log_2(K)$ scores. Furthermore, as the overall complexity of the FIM is governed by the choice of $a(s)$, we can further simplify Equation (11). We propose the *natural* activation function $\nu(x)$ to render the second component constant:

$$\nu(x) = \begin{cases} 0 & \text{if } x \leq C - \frac{A}{2} \\ \frac{1 + sin\left(\frac{\pi(x-C)}{A}\right)}{2} & \text{if } C - \frac{A}{2} \leq x \leq C + \frac{A}{2} \\ 1 & \text{if } x \geq C + \frac{A}{2} \end{cases} \tag{12}$$

$C$ is a parameter that can be used to shift the function along the $x$ axis and to modify the categorical probabilities at initializations $-$ when it is reasonable to expect the scores to be distributed around zero. In the experiment we use $C = 0$. Parameter $A$ can be changed to modify the slope of the function around $C$. In the experiments, to have a fair comparison between the natural activation $\nu$ with the sigmoid function $\sigma$ we set $A$ so that $\frac{\partial \nu}{\partial s}\big|_{s=0} = \frac{\partial \sigma}{\partial s}\big|_{s=0}$ resulting in $A = 2\pi$.

We term $\nu$ the *natural* activation function as it simplifies the FIM in a way that aligns with the objectives of natural gradient methods, as demonstrated in Corollary 4.3.

**Corollary 4.3.** *The Fisher Information Matrix $G_a(s)$ for the catnat parameterization using the natural activation function $\nu$ is:*

$$G_\nu(s)_{ij} = \begin{cases} 0 & \text{if } i \neq j \\ P(a_i)\left(\frac{\pi}{A}\right)^2 & \text{if } i = j \text{ and } |s_i - C_i| < \frac{A}{2} \end{cases} \tag{13}$$

*with the value for $i = j$ at $|s_i - C_i| = \frac{A}{2}$ defined by continuity.*

The corollary is proved in Appendix C. The corollary shows that using the *natural* activation eliminates the dependence of each diagonal entry $G_\nu(s)_{ii}$ on the local score $s_i$, leaving only the ancestor-dependent probability term $P(a_i)$.

## 5 EXPERIMENTS

We evaluate three parameterizations for categorical latent random variables: the *softmax* function, the *catnat* parameterization with *sigmoid* activation, and the *catnat* parameterization with *natural* activation function. The evaluation spans three distinct domains that rely on such variables: Graph Structure Learning (GSL), Variational Autoencoders (VAE), and Reinforcement Learning (RL). These domains allow to assess the proposed method under diverse conditions, varying factors such as the gradient estimator employed for the latent distribution parameters, the number of categories ($K$), the number of latent variables ($N$), the form of the loss or reward function and the downstream task considered. Empirical results show that both hierarchical parameterizations typically converge to better optima, with the proposed *natural* activation function yielding superior performance in the majority of the cases.

### 5.1 GRAPH STRUCTURE LEARNING

Graph Neural Networks (GNNs) (Scarselli et al., 2008) are a class of models that leverage relational information, encoded in an adjacency matrix $A$, as an inductive bias to improve performance on various predictive tasks (Fout et al., 2017; Shlomi et al., 2020). Often, the optimal graph structure is not available and must be inferred from the data, a process known as Graph Structure Learning (GSL) (Kipf et al., 2018; Franceschi et al., 2019; Fatemi et al., 2021). In this context, the adjacency matrix $A$ is frequently treated as a collection of latent categorical random variables $\vec{C}$, where a Bernoulli random variable typically models the existence of each edge (Franceschi et al.,

Table 1: Different datasets are generated with different latent distributions. The latent distribution is determined by the Bernoulli probability $\theta^*$ of sampling edges from communities as in Figure 5.

| True Bernoulli probability $\theta^*$ | Binary entropy per edge (shannons) |
|---|---|
| 0.1 | 0.47 |
| 0.25 | 0.81 |
| 0.5 | 1 |
| 0.75 | 0.81 |
| 0.9 | 0.47 |

Table 2: Test metrics of models trained on datasets generated with different true latent parameters $\theta^*$. ES, PP-MAE, and PP-MSE measure predictive performance, while MAE in $\theta$ evaluates calibration on the latent distribution parameters. For all metrics, lower values indicate better performance. **Bold** numbers indicate the best-performing models. Optimal values are estimated using the true data-generating model.

| $\theta^*$ | Activation | ES loss | PP-MAE | PP-MSE | MAE on $\theta$ |
|---|---|---|---|---|---|
| 0.1 | *sigmoid* | $7.445 \pm 0.008$ | $0.3843 \pm 0.0006$ | $0.621 \pm 0.001$ | $0.0077 \pm 0.0002$ |
| | *natural* | $\mathbf{7.429 \pm 0.011}$ | $\mathbf{0.3842 \pm 0.0006}$ | $\mathbf{0.618 \pm 0.002}$ | $\mathbf{0.0054 \pm 0.0002}$ |
| | Optimal value | $7.417 \pm 0.008$ | $0.3844 \pm 0.0007$ | $0.615 \pm 0.001$ | $0$ |
| 0.25 | *sigmoid* | $10.892 \pm 0.012$ | $0.8228 \pm 0.0009$ | $1.312 \pm 0.003$ | $0.0080 \pm 0.0001$ |
| | *natural* | $\mathbf{10.872 \pm 0.017}$ | $\mathbf{0.8213 \pm 0.0012}$ | $\mathbf{1.307 \pm 0.004}$ | $\mathbf{0.0060 \pm 0.0004}$ |
| | Optimal value | $10.848 \pm 0.009$ | $0.8199 \pm 0.0007$ | $1.300 \pm 0.002$ | $0$ |
| 0.5 | *sigmoid* | $14.955 \pm 0.007$ | $1.2536 \pm 0.0006$ | $2.469 \pm 0.002$ | $0.0191 \pm 0.0005$ |
| | *natural* | $\mathbf{14.942 \pm 0.028}$ | $\mathbf{1.2534 \pm 0.0024}$ | $\mathbf{2.468 \pm 0.009}$ | $\mathbf{0.0064 \pm 0.0005}$ |
| | Optimal value | $14.923 \pm 0.015$ | $1.2522 \pm 0.0015$ | $2.461 \pm 0.005$ | $0$ |
| 0.75 | *sigmoid* | $10.716 \pm 0.043$ | $0.8017 \pm 0.0032$ | $1.273 \pm 0.010$ | $0.0181 \pm 0.0007$ |
| | *natural* | $\mathbf{10.672 \pm 0.012}$ | $\mathbf{0.7983 \pm 0.0011}$ | $\mathbf{1.267 \pm 0.003}$ | $\mathbf{0.0045 \pm 0.0002}$ |
| | Optimal value | $10.671 \pm 0.014$ | $0.7942 \pm 0.0010$ | $1.267 \pm 0.003$ | $0$ |
| 0.9 | *sigmoid* | $7.377 \pm 0.018$ | $0.4265 \pm 0.0015$ | $0.614 \pm 0.003$ | $0.0145 \pm 0.0003$ |
| | *natural* | $\mathbf{7.353 \pm 0.013}$ | $\mathbf{0.4026 \pm 0.0014}$ | $\mathbf{0.611 \pm 0.002}$ | $\mathbf{0.0036 \pm 0.0003}$ |
| | Optimal value | $7.341 \pm 0.011$ | $0.3840 \pm 0.0008$ | $0.611 \pm 0.002$ | $0$ |

2019; Elinas et al., 2020; Zambon et al., 2023; Cini et al., 2023; Manenti et al., 2025). We adopt the experimental setup from Manenti et al. (2025), generating synthetic data with a Graph Neural Network (GNN), $f_{\psi^*}(x, A)$. This GNN computes an output $y^*$ from random input features $x$ and a latent graph $A$, which we sample from a multivariate Bernoulli distribution, $P_{\theta^*}(A)$. The ground-truth parameters $\theta_{ij}^*$ are set to the same non-zero value $\theta^*$ for edges forming the community structure depicted in Figure 5 and are zero otherwise. We use this dataset to train a model with an identical architecture to recover the underlying graph structure and GNN parameters. We optimize the model using the Energy Score (ES) (Gneiting & Raftery, 2007) loss for its calibration advantages (Manenti et al., 2025). We use the score function gradient estimator (Williams, 1992) with the LOO baseline to train the latent parameters. We provide additional details in Appendix D. As each categorical random variable is bivariate, the resulting hierarchical parameterization has a depth of one. We therefore compare the *natural* activation function proposed herein with the standard *sigmoid*.

To compare the score parameterizations under different entropy settings we generate five datasets with different true latent parameters $\theta^*$. Experiment configurations are detailed in Table 1. The task in this setting is twofold: (i) to make optimal point predictions, measured for example by the Point Prediction Mean Absolute Error (PP-MAE) and Mean Squared Error (PP-MSE), and (ii) to learn the correct graph structure, i.e., to accurately estimate the true parameters $\theta^*$. The latter is evaluated, for example, by the mean absolute error on the distribution parameters (MAE on $\theta$), $\langle |\theta_{ij} - \theta_{ij}^*| \rangle$.

The experimental results in Table 2 show that the *natural* activation $\nu$ consistently outperforms the standard *sigmoid* $\sigma$ across all metrics and data-generating conditions, with the largest gains in learning the underlying latent distribution. In particular, the natural parameterization recovers the true data-generating parameters more accurately, as measured by the MAE on $\theta$. This improvement is especially pronounced in the highest-entropy setting ($\theta^* = 0.5$), where the error is reduced by nearly a factor of three. In terms of predictive performance, the natural parameterization also achieves lower mean scores on the ES loss and both point prediction errors.

## 5.2 CATEGORICAL VAE

Variational autoencoders (VAEs) (Kingma & Welling, 2014) constitute a class of deep generative models that learn compact latent representations of data via a probabilistic framework. The original VAE framework employs a continuous latent distribution—typically Gaussian—which may not suit

Table 3: Test set negative log likelihood on the MNIST dataset. Negative log-likelihoods are estimated with 512 importance samples (Burda et al., 2016) (lower is better). Models are compared across the number of categorical variables $N$, categories $K$, and categorical parameterizations. **Bold** denotes the best-performing model for each $(N, K)$ setting, and underline the second best.

| $N$ | Param. | MNIST | | | Binary MNIST | | |
|---|---|---|---|---|---|---|---|
| | | $K = 8$ | $K = 16$ | $K = 32$ | $K = 8$ | $K = 16$ | $K = 32$ |
| | *softmax* | $100.9 \pm 0.5$ | $98.1 \pm 0.7$ | $98.6 \pm 0.7$ | $84.9 \pm 0.8$ | $81.0 \pm 1.2$ | $79.9 \pm 0.5$ |
| 10 | *catnat* $\sigma$ | $\mathbf{99.5 \pm 0.2}$ | $\underline{97.7 \pm 0.4}$ | $\mathbf{96.6 \pm 0.2}$ | $\mathbf{83.0 \pm 0.6}$ | $\underline{78.8 \pm 0.6}$ | $\mathbf{76.9 \pm 0.7}$ |
| | *catnat* $\nu$ | $\underline{99.8 \pm 0.4}$ | $\mathbf{97.6 \pm 0.2}$ | $\underline{96.9 \pm 0.4}$ | $\underline{83.2 \pm 0.5}$ | $\mathbf{78.7 \pm 0.3}$ | $77.3 \pm 0.4$ |
| | *softmax* | $97.8 \pm 0.2$ | $97.5 \pm 0.5$ | $98.2 \pm 0.8$ | $78.3 \pm 0.5$ | $78.1 \pm 0.4$ | $79.2 \pm 1.0$ |
| 20 | *catnat* $\sigma$ | $\mathbf{97.5 \pm 0.3}$ | $\mathbf{96.9 \pm 0.4}$ | $\underline{97.0 \pm 0.3}$ | $\underline{77.5 \pm 1.1}$ | $76.7 \pm 0.7$ | $\mathbf{76.2 \pm 0.5}$ |
| | *catnat* $\nu$ | $\underline{97.7 \pm 0.2}$ | $\underline{97.0 \pm 0.4}$ | $\mathbf{96.9 \pm 0.4}$ | $\mathbf{77.1 \pm 0.4}$ | $\mathbf{76.6 \pm 0.3}$ | $\underline{76.8 \pm 0.4}$ |
| | *softmax* | $98.8 \pm 0.7$ | $98.8 \pm 0.9$ | $99.3 \pm 0.7$ | $79.0 \pm 0.5$ | $79.2 \pm 0.9$ | $80.6 \pm 0.6$ |
| 30 | *catnat* $\sigma$ | $\underline{98.1 \pm 0.4}$ | $\underline{97.6 \pm 0.4}$ | $\underline{97.9 \pm 0.5}$ | $\underline{77.9 \pm 0.7}$ | $\underline{77.8 \pm 0.6}$ | $\mathbf{77.9 \pm 1.1}$ |
| | *catnat* $\nu$ | $\mathbf{97.9 \pm 0.3}$ | $\mathbf{97.6 \pm 0.3}$ | $\mathbf{97.7 \pm 0.8}$ | $\mathbf{77.9 \pm 0.6}$ | $\mathbf{77.7 \pm 0.5}$ | $\underline{78.0 \pm 0.7}$ |

data with inherently discrete factors (Van Den Oord et al., 2017). To address this limitation, VAEs have been extended to incorporate categorical variables $\vec{C}$, enabling more accurate modeling of such structures (Jang et al., 2017; Maddison et al., 2017). In this configuration, the latent space is parameterized by one or more categorical random variables.

We trained a variational autoencoder (VAE) with a discrete, categorical latent space[2] on the MNIST dataset (LeCun & Cortes, 2010) and on a binarized version of the same dataset, where pixels are thresholded at $0.5$ of their maximum intensity value (Akrami et al., 2022). Its encoder, a Convolutional Neural Network, processes an input image $x$ to produce a tensor of unnormalized scores, $\vec{s} \in \mathbb{R}^{N \times K}$, which parameterize the approximate posterior distribution $q(\vec{C}|x)$. The latent space is structured as a composite of $N$ independent categorical variables, where each variable can assume one of $K$ distinct classes. To enable gradient flow through the discrete sampling process, we employ the Gumbel-Softmax trick (Jang et al., 2017), which generates differentiable sample tensors $\vec{C} \in [0, 1]^{N \times K}$. Note that the samples used by the Gumbel-Softmax trick are continuous relaxations of one-hot vectors. The latent samples $\vec{C}$ are then fed into the decoder, a corresponding transposed convolutional network (Zeiler et al., 2010), to generate a reconstructed image $\hat{y}$.

To form the categorical distributions from the encoder's scores, we test three parameterization schemes: the *softmax* function, the *catnat* function with *sigmoid* activation function and the *catnat* function with *natural* activation. The model's training objective is the minimization of the Evidence Lower Bound (ELBO) (Kingma & Welling, 2014). We defer additional experimental details to Appendix E.

The results in Table 3 show a clear performance advantage for both *catnat* parameterizations over the *softmax* across all experiments. This indicates that the benefits of the proposed parameterization are relevant in the probabilistic generative modeling case. Importantly, these improvements are observed across a wide range of latent configurations ($N \in \{10, 20, 30\}$, $K \in \{8, 16, 32\}$), underscoring the robustness of the approach to changes in model capacity and latent space complexity. Within the hierarchical methods, the natural activation function $\nu$ yields a slight improvement over the sigmoid function $\sigma$ in the majority of the settings, although the two are statistically equivalent on average.

Overall, these results demonstrate that in practical real-world scenarios, simply replacing the standard *softmax* with the proposed *catnat* parameterization facilitates training and consistently improves downstream performance.

## 5.3 REINFORCEMENT LEARNING

Reinforcement learning (RL) is a framework where an agent learns to make sequential decisions by interacting with an environment in order to maximize a cumulative reward signal (Sutton et al.,

---

[2]Our implementation is based on the code available at https://github.com/jxmorris12/categorical-vae.

1998). In policy-based approaches (Sutton et al., 1998), the agent's strategy is directly parameterized by a policy $\pi$, which maps observed states to actions. In many domains, such as board games or classic Atari video games (Bellemare et al., 2013; Mnih et al., 2013; 2015), the action space is discrete, requiring the agent to select from a finite set of choices at each step. In such settings, the policy produces a categorical distribution over the available actions.

In this setting, we employ the Proximal Policy Optimization (PPO) algorithm (Schulman et al., 2017) on the discrete-action Atari environments Breakout and Seaquest (Mnih et al., 2013; 2015). We adopt the PPO implementation from Huang et al. (2022) in which an agent uses a shared-parameter actor-critic architecture, with a deep convolutional network processing stacked game frames to produce a latent state representation. This state is then fed into two separate heads: a value head that estimates the state-value function, and a policy head that outputs scores for the action distribution. Additional experimental details are provided in Appendix F

Due to the computational burden of these experiments, an exhaustive hyperparameter search was not feasible. Instead, for each method and environment, we selected promising configurations by sampling 160 trials with a Tree-structured Parzen Estimator (TPE) Bayesian sampler (Bergstra et al., 2011). The top 10 resulting configurations were then re-evaluated across 10 independent random seeds to gather performance statistics. Within this framework, we tested two methods to convert the policy head's scores into action probabilities: the standard *softmax* function and the *catnat* using the *natural* activation function.

Table 4 reports the final episodic returns on Seaquest and Breakout environments. The *catnat* parameterization yields better performance w.r.t. the standard *softmax* function, with a modest improvement in Breakout and a more substantial gain in the more complex Seaquest environment. These results indicate that the information-geometric properties of *catnat* translate into practical benefits even in high-dimensional, sequential decision-making tasks.

Table 4: Final episodic returns on Seaquest and Breakout environments. The higher the better.

| Parameterization | RL Environment | |
|---|---|---|
| | Breakout | Seaquest |
| *softmax* | $398 \pm 25$ | $1875 \pm 312$ |
| *catnat* $\nu$ | $\mathbf{406 \pm 34}$ | $\mathbf{2164 \pm 533}$ |

A notable characteristic of these experiments is the high variance, reflected in the large standard deviations relative to the mean returns. This variance is typical in deep RL due to sensitivity to initialization, stochasticity in exploration, and non-stationarity of training dynamics. The fact that *catnat* maintains a consistent performance advantage despite this variance suggests that its benefits are robust rather than artifacts of specific hyperparameter settings. In particular, the larger relative gains in Seaquest, where the action space is richer and exploration dynamics more complex, point to potential advantages in environments with increased complexity. A more exhaustive search could provide a clearer picture of the potential performance ceiling of *catnat*, with future work investigating how the relative benefits of this parameterization scale with task difficulty, action space size, or agent capacity.

## 6  CONCLUSIONS

We introduced a new perspective for improving training of models with latent categorical random variables. Specifically, we showed that replacing the standard *softmax* parameterization with a *catnat* function — a hierarchical sequence of binary decisions — yields favorable information-geometric properties. Empirical results across diverse settings indicate that these properties facilitate gradient-based optimization and provide better parameter configurations leading to improved final performance. Two main directions for future work remain. First, although our study focused on categorical distributions, the findings suggest that parameterizations that induce a diagonal Fisher Information Matrix consistently improve performance. Extending this approach to other families of continuous and discrete distributions is a promising avenue for future research. Second, our experiments were not designed to engineer models for state-of-the-art performance, but rather to demonstrate the broad applicability and effectiveness of the proposed approach. Nonetheless, application-specific state-of-the-art methods that rely on categorical random variables are likely to benefit from the *catnat* parameterization, and could thereby achieve new state-of-the-art results with minimal effort.

## REPRODUCIBILITY STATEMENT

We have taken several measures to ensure the reproducibility of our results. The proposed *catnat* parameterization is simple to implement, and its full construction is provided in Section 4.2. All theoretical results are formally stated in the main text and rigorously proved in the appendices (see Appendix A, B, and C). The experimental settings rely on publicly available codebases: Graph Structure Learning builds on Manenti et al. (2025), the categorical VAE implementation follows this open-source repository, and the reinforcement learning experiments use the high-quality PPO implementation from Huang et al. (2022). For each experiment, the datasets used in our experiments are all standard and publicly available: the GSL dataset is generated following the procedure in Manenti et al. (2025), the VAE experiments use the MNIST and binarized MNIST datasets (LeCun & Cortes, 2010; Akrami et al., 2022), and the RL tasks are based on Atari environments provided by the Gymnasium library (Bellemare et al., 2013; Towers et al., 2024). Furthermore, we provide detailed descriptions of the model architectures, hyperparameter searches, optimization strategies, and evaluation metrics in Appendix D, Appendix E, and Appendix F, respectively. Together, these elements provide sufficient information to fully reproduce our theoretical and empirical results.

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

## APPENDIX

## A  PROOF OF PROPOSITION 4.1

Here we prove that the Fisher Information Matrix for a categorical distribution parametrized by a *softmax* $\left(\text{i.e., } p_i = \frac{e^{s_i}}{\sum_{k=1}^{K} e^{s_k}}\right)$ is:

$$
G_{softmax}(s) = \begin{bmatrix} p_1(1-p_1) & -p_1 p_2 & \cdots & -p_1 p_K \\ -p_2 p_1 & p_2(1-p_2) & \cdots & -p_2 p_K \\ \vdots & \vdots & \ddots & \vdots \\ -p_K p_1 & -p_K p_2 & \cdots & p_K(1-p_K) \end{bmatrix}
$$

*Proof.* For a single observation take $C = (C_1, \ldots, C_K)$ be a one-hot encoded vector with $C_{\bar{k}} = 1$ if the observed category is $\bar{k}$ and zero elsewhere. The log-likelihood is:

$$log(p(C|s)) = \sum_{k=1}^{K} C_k log(p_k) = \sum_{k=1}^{K} C_k \left( s_k - log \left( \sum_{k'=1}^{K} e^{s_{k'}} \right) \right) \tag{14}$$

We have:

$$\frac{\partial log(p(C_k = 1|s))}{\partial s_i} = \delta_{ki} - p_i \tag{15}$$

With $\delta_{ki}$ being the Kronecker delta.

The Fisher Information Matrix is:

$$G_{softmax}(s)_{ij} = \mathbb{E}_{C \sim p(C|s)} \left[ \frac{\partial log(p(C|s))}{\partial s_i} \frac{\partial log(p(C|s))}{\partial s_j} \right] \tag{16}$$

$$= \sum_{k=1}^{K} p_k (\delta_{ki} - p_i)(\delta_{kj} - p_j) \tag{17}$$

- For diagonal elements:

$$G_{softmax}(s)_{ii} = \sum_{k=1}^{K} p_k (\delta_{ki} - p_i)^2 = p_i(1 - p_i) \tag{18}$$

- For off-diagonal elements:

$$G_{softmax}(s)_{ij} = \sum_{k=1}^{K} p_k (\delta_{ki} - p_i)(\delta_{kj} - p_j) \tag{19}$$

$$= \sum_{k=1}^{K} p_k (\delta_{ki}\delta_{kj} - \delta_{ki}p_j - \delta_{kj}p_i + p_i p_j) \tag{20}$$

$$= -p_i p_j \tag{21}$$

yielding the Fisher Information Matrix $G_{softmax}(s)$ stated in the proposition. $\square$

## B  PROOF OF THEOREM 4.2

To prove Theorem 4.2, we use the following series of lemmas and propositions.

**Lemma B.1.** *Given a set of bits $[b_1, ..., b_H]$ then $a_{\boxed{ID}}$ is an ancestor of $p_{b_1,...,b_H}$ if and only if $\boxed{ID} = \boxed{b_1, ..., b_{h-1}}$*

*Proof.* By construction the binary number $[b_1, ..., b_H]$ in $p_{b_1,...,b_H}$ represents the binary decisions taken at each hierarchy level $h$. In particular, the first $h - 1$ terms $[b_1, ..., b_{h-1}]$ represent, in order, the first $h - 1$ binary decisions.

For each hierarchy level $h$, each node $a_{\boxed{ID}}$ is identified by $\boxed{ID}$. The numerical value represents its position reading right to left by construction (i.e., $b_h = 1$ corresponds to descending left). The first bit in $\boxed{ID}$ splits the $2^{h-1}$ numbers in half (i.e., for the left half the first bit is one, for the right half is zero). The subsequent bits recursively split the selected group in half following the same logic. Thus, each bit in $\boxed{ID}$ can be viewed as a binary decision of moving left or right. Thus, $a_{\boxed{b_1, ..., b_{h-1}}}$ is the node reached from the root following binary decisions $b_1, ..., b_{h-1}$. Since each $p$ has $H$ ancestors and the root is common the lemma is proved.

$\square$

**Lemma B.2.** *Given $a_\alpha$ and $p_\gamma$,*

$$\text{if } p_\gamma \text{ is not a descendant of } a_\alpha \implies \frac{\partial}{\partial a_\alpha} log(p_\gamma) = 0. \tag{22}$$

*Proof.* Consider the binary representation of $a_\alpha$. From Lemma B.1 $a_{\boxed{\text{ID}}}$ is not an ancestor of $p_{b_1,\dots,b_H}$ then $\boxed{\text{ID}} \neq \boxed{b_1,\dots,b_{h-1}}$. In that case $a_{\boxed{\text{ID}}}$ is not a term in (8) and thus $\frac{\partial}{\partial a_\alpha} log(p_\gamma) = \frac{1}{p_\gamma} \frac{\partial}{\partial a_\alpha} p_\gamma = 0$. $\qquad\square$

**Corollary B.3.** *Given $a_\alpha$, $a_\beta$ and $p_\gamma$ with $\alpha \neq \beta$,*

*If $a_\alpha$ is neither a descendant nor an ancestor of $a_\beta$ $\implies$ $\frac{\partial}{\partial a_\alpha} log(p_\gamma) \frac{\partial}{\partial a_\beta} log(p_\gamma) = 0$.*

*Proof.* If $a_\alpha$ is neither a descendant nor an ancestor of $a_\beta$ then they do not share any descendant and thus for Lemma B.2 the Corollary is trivially proved. $\qquad\square$

**Proposition B.4.** *The Fisher Information Matrix $G_a(s)$ for the catnat parameterization is diagonal.*

*Proof.* To prove the Proposition we prove that all the off-diagonal terms of $G_a(s)$ are zero. By definition:

$$G_a(s)_{\alpha\beta} = \mathbb{E}_{C \sim p(C|s)} \left[ \frac{\partial}{\partial s_\alpha} log(p(C|s)) \frac{\partial}{\partial s_\beta} log(p(C|s)) \right]$$

$$= \mathbb{E}_{C \sim p(C|s)} \left[ \frac{\partial a_\alpha}{\partial s_\alpha} \frac{\partial}{\partial a_\alpha} log(p(C|s)) \frac{\partial a_\beta}{\partial s_\beta} \frac{\partial}{\partial a_\beta} log(p(C|s)) \right]$$

$$= \sum_{\vec{b} \in \{0,1\}^H} p_{\vec{b}} \left[ \frac{\partial a_\alpha}{\partial s_\alpha} \frac{\partial}{\partial a_\alpha} log(p_{\vec{b}}) \frac{\partial a_\beta}{\partial s_\beta} \frac{\partial}{\partial a_\beta} log(p_{\vec{b}}) \right] \tag{23}$$

From Corollary B.3 if $a_\alpha$ is neither a descendant nor an ancestor of $a_\beta$ the term in the square brackets is zero. We thus consider $a_\alpha$ being an ancestor of $a_\beta$, since the FIM is symmetric this is not restrictive. From Lemma B.2 the only terms that may produce nonzero addends are from the $\vec{b}$ that are descendant $\mathcal{D}_\beta$ of $a_\beta$. Thus:

$$G_a(s)_{\alpha\beta} = \sum_{\vec{b} \in \mathcal{D}_\beta} p_{\vec{b}} \left[ \frac{\partial a_\alpha}{\partial s_\alpha} \frac{\partial}{\partial a_\alpha} log(p_{\vec{b}}) \frac{\partial a_\beta}{\partial s_\beta} \frac{\partial}{\partial a_\beta} log(p_{\vec{b}}) \right] \tag{24}$$

We call $h_\alpha$ and $h_\beta$ the hierarchies of $a_\alpha$ and $a_\beta$ and consider their binary representation $a_{\boxed{\text{ID}}_\alpha}$ and $a_{\boxed{\text{ID}}_\beta}$. From Equation 8 we write the log-likelihood derivative as:

$$\frac{\partial}{\partial a_{\boxed{\text{ID}}}} log\left( p_{b_1,\dots,b_H} \right) = \mathbb{1}\left[ [b_1,\dots,b_{h-1}] = \boxed{\text{ID}} \right] \frac{2b_h - 1}{a_{\boxed{\text{ID}}}^{b_h} (1 - a_{\boxed{\text{ID}}})^{(1-b_h)}} \tag{25}$$

where $\mathbb{1}[\cdot]$ is the indicator function, which evaluates to 1 if the given condition is true and 0 if it is false. The $\vec{b} \in \mathcal{D}_\beta$ share the same first $h_\beta - 1$ bits. Since $a_\alpha$ is an ancestor of $a_\beta$ then $h_\beta > h_\alpha$ and thus $b_{h_\alpha}$ is the same for all $\vec{b} \in \mathcal{D}_\beta$. Then:

$$G_a(s)_{\alpha\beta} = K \sum_{\vec{b} \in \mathcal{D}_\beta} p_{\vec{b}} \left[ \frac{\partial}{\partial a_\beta} log(p_{\vec{b}}) \right]$$

$$= K \sum_{\substack{\vec{b} \in \mathcal{D}_\beta \\ b_{h_\beta}=1}} p_{\vec{b}} \underbrace{\left[ \frac{\partial}{\partial a_\beta} log(p_{\vec{b}}) \right]}_{a_{\boxed{\text{ID}}_\beta}^{-1}} + K \sum_{\substack{\vec{b} \in \mathcal{D}_\beta \\ b_{h_\beta}=0}} p_{\vec{b}} \underbrace{\left[ \frac{\partial}{\partial a_\beta} log(p_{\vec{b}}) \right]}_{-(1-a_{\boxed{\text{ID}}_\beta})^{-1}}$$

$$= K \sum_{\substack{\vec{b} \in \mathcal{D}_\beta \\ b_{h_\beta}=1}} \frac{p_{\vec{b}}}{a_{\boxed{\text{ID}}_\beta}} - K \sum_{\substack{\vec{b} \in \mathcal{D}_\beta \\ b_{h_\beta}=0}} \frac{p_{\vec{b}}}{1 - a_{\boxed{\text{ID}}_\beta}}$$

$$= \frac{K}{a_{\boxed{\text{ID}}_\beta}} \sum_{\substack{\vec{b} \in \mathcal{D}_\beta \\ b_{h_\beta}=1}} p_{\vec{b}} - \frac{K}{1 - a_{\boxed{\text{ID}}_\beta}} \sum_{\substack{\vec{b} \in \mathcal{D}_\beta \\ b_{h_\beta}=0}} p_{\vec{b}} \tag{26}$$

By construction:

$$\sum_{\substack{\vec{b}\in\mathcal{D}_\beta \\ b_{h_\beta}=1}} p_{\vec{b}} = P\left(\text{descend to node } a_{\boxed{\text{ID}}_\beta}\right) \cdot a_{\boxed{\text{ID}}_\beta}$$

$$\sum_{\substack{\vec{b}\in\mathcal{D}_\beta \\ b_{h_\beta}=0}} p_{\vec{b}} = P\left(\text{descend to node } a_{\boxed{\text{ID}}_\beta}\right) \cdot (1 - a_{\boxed{\text{ID}}_\beta})$$

Thus, $G_a(s)_{\alpha\beta} = 0$ for off-diagonal terms. $\qquad\square$

**Proposition B.5.** *The diagonal terms of the Fisher Information Matrix $G_a(s)$ for the catnat parameterization are:*

$$G_a(s)_{ii} = P(a_i)\left(\frac{\partial a_i}{\partial s_i}\right)^2 \left(\frac{1}{a_i(1-a_i)}\right) \tag{27}$$

*Proof.* Reusing arguments from the previous proofs we can write:

$$G_a(s)_{ii} = \mathbb{E}_{C\sim p(C|s)}\left[\left(\frac{\partial}{\partial s_i}log(p(C|s))\right)^2\right]$$

$$= \sum_{\vec{b}\in\mathcal{D}_i} p_{\vec{b}}\left(\frac{\partial}{\partial s_i}log(p_{\vec{b}})\right)^2$$

$$= \sum_{\vec{b}\in\mathcal{D}_i} p_{\vec{b}}\left(\frac{\partial a_i}{\partial s_i}\frac{\partial}{\partial a_i}log(p_{\vec{b}})\right)^2 \tag{28}$$

$$= \left(\frac{\partial a_i}{\partial s_i}\right)^2 \sum_{\vec{b}\in\mathcal{D}_i} p_{\vec{b}}\left(\frac{\partial}{\partial a_i}log(p_{\vec{b}})\right)^2$$

$$= \left(\frac{\partial a_i}{\partial s_i}\right)^2 \left[\sum_{\substack{\vec{b}\in\mathcal{D}_i \\ b_{h_i}=1}} p_{\vec{b}}\left(\frac{1}{a_{\boxed{\text{ID}}_i}}\right)^2 + \sum_{\substack{\vec{b}\in\mathcal{D}_i \\ b_{h_i}=0}} p_{\vec{b}}\left(\frac{-1}{1-a_{\boxed{\text{ID}}_i}}\right)^2\right]$$

$$= \left(\frac{\partial a_i}{\partial s_i}\right)^2 \left[P(a_i)\left(\frac{1}{a_i}\right)^{\cancel{2}}\cancel{a_i} + P(a_i)\left(\frac{1}{1-a_i}\right)^{\cancel{2}}\cancel{(1-a_i)}\right]$$

$$= P(a_i)\left(\frac{\partial a_i}{\partial s_i}\right)^2 \left(\frac{1}{a_i(1-a_i)}\right)$$

$\qquad\square$

Theorem 4.2 follows naturally from Proposition B.4 and Proposition B.5.

## C  PROOF OF COROLLARY 4.3

*Proof.* To prove the corollary we start from Theorem 4.2 and substitute the definition in (12).

For the natural activation function $\nu$:

$$
\left(\frac{\partial \nu_i}{\partial s_i}\right)^2 \left(\frac{1}{\nu_i(1-\nu_i)}\right) = \left(\frac{\partial}{\partial s_i} \frac{1 + sin\left(\frac{\pi(s_i-C)}{A}\right)}{2}\right)^2 \left(\frac{1}{\left(\frac{1+sin\left(\frac{\pi(s_i-C)}{A}\right)}{2}\right)\left(1 - \frac{1+sin\left(\frac{\pi(s_i-C)}{A}\right)}{2}\right)}\right)
$$

$$
= \left(\frac{\pi}{2A}cos\left(\frac{\pi(s_i-C)}{A}\right)\right)^2 \left(\frac{4}{\left(1 + sin\left(\frac{\pi(s_i-C)}{A}\right)\right)\left(1 - sin\left(\frac{\pi(s_i-C)}{A}\right)\right)}\right)
$$

$$
= \left(\frac{\pi}{2A}cos\left(\frac{\pi(s_i-C)}{A}\right)\right)^2 \left(\frac{4}{\left(1 - sin^2\left(\frac{\pi(s_i-C)}{A}\right)\right)}\right)
$$

$$
= \left(\frac{\pi}{2A}cos\left(\frac{\pi(s_i-C)}{A}\right)\right)^2 \left(\frac{4}{cos^2\left(\frac{\pi(s_i-C)}{A}\right)}\right)
$$

$$
= \frac{\pi^2}{A^2}
$$

Thus,

$$
G_\nu(s)_{ii} = P(a_i) \frac{\pi^2}{A^2} \tag{29}
$$

$\square$

# D EXPERIMENTAL DETAILS: GRAPH STRUCTURE LEARNING

This appendix summarizes the experimental setup for the Graph Structure Learning experiment, adapted from Manenti et al. (2025).

## D.1 DATA-GENERATING PROCESS

The dataset is generated from a system model comprising two components: a latent graph distribution $P_A^{\theta^*}$ that produces a random adjacency matrix $A$, and a Graph Neural Network $f_{\psi^*}$ that maps an input feature matrix $x$ and the graph $A$ to an output $y$.

### LATENT GRAPH DISTRIBUTION

The latent graph structure $A$ is sampled from a multivariate Bernoulli distribution parameterized by a matrix of probabilities $\theta_{ij}^*$:

$$
P_{\theta^*}(A) = \prod_{i,j} (\theta_{ij}^*)^{A_{ij}} (1 - \theta_{ij}^*)^{1-A_{ij}} \tag{30}
$$

Each entry $A_{ij}$ represents a potential edge, sampled independently with a success probability of $\theta_{ij}^*$. The ground-truth parameters $\theta_{ij}^*$ are set to the same non-zero value $\theta^*$ for edges forming the community structure depicted in Figure 5 and are zero otherwise. For the experiments we use a graph with 4 communities.

### GNN ARCHITECTURE

The GNN function $f_{\psi^*}$ used to processes the sampled graph $A$ and a random input feature matrix $x \in \mathbb{R}^{N \times d_{in}}$ is a GCN (Kipf & Welling, 2017). The input features are sampled from a normal distribution, $x \sim \mathcal{N}(0, \sigma_x^2 \mathbb{I})$ with $\sigma_x = 1$.

This generation process yielded a dataset of 10,000 input-output $(x, y)$ pairs, partitioned into a training set (80%), a validation set (10%), and a test set (10%). The learnable model we train has an identical architecture to the one described above.

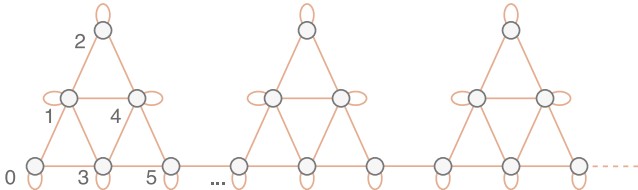

Figure 5: The base graph structure used to generate adjacency matrices for the experiments in Section 5.1. The matrices are sampled as subgraphs from this structure, where each orange edge is included with an independent probability of $\theta_{ij}^*$, according to the distribution $P_{\theta^*}(A)$ in (30). Image taken from Manenti et al. (2025)

## D.2 LEARNABLE MODEL

For the learnable deep learning architecture, we employ the same class of latent graph distribution and GNN architecture as in the data-generating model. We jointly learn the parameters of the latent graph distribution, denoted by $\theta$, and the parameters of the GNN, denoted by $\psi$.

We perform a two-stage grid search over learning rates. In all experiments, the learning rate for the GNN parameters $\psi$ and the latent graph parameters $\theta$ is chosen from the same grid. The search procedure is:

1. Coarse grid: Test rates in $\{0.005, 0.01, 0.02, 0.05, 0.1, 0.2, 0.5\}$.

2. Refined grid: Centered around the best-performing coarse rate (selected by validation loss). The refined grids used in our experiments were:

   - Sigmoid parameterization: $\{0.025, 0.03, 0.037, 0.045, 0.055, 0.067, 0.082\}$.
   - Natural parameterization: $\{0.012, 0.015, 0.018, 0.022, 0.027, 0.033, 0.041\}$.

The final learning rate for each run is the one that yields the lowest validation loss. Using this best learning rate, we train 10 models to compute aggregate statistics.

## D.3 SCORE FUNCTION GRADIENT ESTIMATOR

To compute gradients with respect to the parameters $\theta_{ij}$ of the latent graph distribution, we use the Score Function Gradient Estimator (SFGE), also known as REINFORCE (Williams, 1992; Mohamed et al., 2020). The SFGE allows us to estimate the gradient of an expectation of a function $\mathcal{L}(A)$ as follows:

$$\nabla_\theta \mathbb{E}_{A \sim P_\theta(A)}[\mathcal{L}(A)] = \mathbb{E}_{A \sim P_\theta(A)}[\mathcal{L}(A)\nabla_\theta \log P_\theta(A)]$$

A known issue with the SFGE is its high variance (Mohamed et al., 2020). To mitigate this, we incorporate a baseline term, which reduces variance without introducing bias into the gradient estimate. The gradient is then computed as:

$$\nabla_\theta \mathbb{E}_{A \sim P_\theta(A)}[(\mathcal{L}(A) - b)\nabla_\theta \log P_\theta(A)]$$

We use a multi-sample baseline where, for each sample in a batch of $M$ sampled graphs, the baseline $b$ is constructed using the estimate of the loss from the other $M - 1$ samples.

## D.4 LOSS FUNCTION

The model is trained to learn both the GNN parameters $\psi$ and the graph distribution parameters $\theta$ by minimizing the Energy Score (ES) (Gneiting & Raftery, 2007). The ES is a multivariate extension of the Continuous Ranked Probability Score (CRPS), a proper scoring rule (Matheson & Winkler, 1976) that quantifies the compatibility between the model's predictive distribution and the ground-truth observation $y$.

Given $M$ adjacency matrices $\{A_m\}_{m=1}^M$ sampled from the latent graph distribution $P_\theta(A)$, the empirical ES loss is defined as:

$$\mathcal{L}_{\text{ES}} = \frac{1}{M} \sum_{m=1}^M \|f_\psi(x, A_m) - y\|_2 - \frac{1}{2M(M-1)} \sum_{m \neq n} \|f_\psi(x, A_m) - f_\psi(x, A_n)\|_2$$

### D.5 ADDITIONAL TRAINING PARAMETERS

All models are implemented in PyTorch (Paszke et al., 2017) and trained with the Adam optimizer (Kingma & Ba, 2015). We use a Weight decay of $0$, a batch size of $64$, $M$ equal to $32$ and $40$ epochs per run. Scores were initialized so that $\theta_{ij} \sim \mathcal{U}(0, 0.1)$.

## E EXPERIMENTAL DETAILS: VAE

This appendix summarizes the setup for the experiments with Variational Autoencoders, which we adopt from the code available at the following link: https://github.com/jxmorris12/categorical-vae.

### E.1 MODEL ARCHITECTURE

The Variational Autoencoder (VAE) is composed of an encoder network $q_{\vec{s}}(\vec{C}|x)$ and a decoder network $f_\psi(\hat{y}|\vec{C})$. Both networks are implemented with a convolutional structure.

#### ENCODER & CATEGORICAL LATENT DISTRIBUTION

The encoder processes an input image $x \in \mathbb{R}^{28 \times 28}$, computes a tensor of scores $\vec{s} \in \mathbb{R}^{N \times K}$ — where $N$ is the number of latent categorical variables, and $K$ is the number of classes for each variable — and outputs the latent probabilities $q_{\vec{s}}(\vec{C}|x)$. The default convolutional architecture consists of 3 convolutional layers with ReLU activations, followed by 2 fully-connected layers.

Thus, the latent space is defined by $N$ independent categorical random variables, with each variable taking one of $K$ discrete states. The scores $\vec{s} \in \mathbb{R}^{N \times K}$ computed by the encoder are transformed into latent probabilities $q_{\vec{s}}(\vec{C}|x)$ with different parameterizations. We test three schemes for this parameterization: the *softmax* function, the *catnat* parameterization with *sigmoid* activation function and the *catnat* parameterization with *natural* activation function.

#### DECODER

The decoder takes a set of one-hot latent samples (one vector for each categorical distribution) $\vec{C}$ and processes it through 2 fully-connected layers and 3 transposed convolutional layers to reconstruct an image. A sigmoid activation function in the final layer ensures the output values are bounded within $[0, 1]$. Thus, the decoder $f_\psi(\hat{y}|\vec{C})$ produces a reconstruction whose entries can be interpreted as independent Bernoulli distributions over each pixel.

### E.2 GUMBEL-SOFTMAX REPARAMETERIZATION

To maintain a differentiable computation graph, we use the Gumbel-Softmax trick (Jang et al., 2017) to approximate sampling from $q_{\vec{s}}(\vec{C}|x)$. The temperature hyperparameter $\tau$ controls the smoothness of the approximation; as $\tau \to 0$, the samples converge to discrete one-hot vectors. During training, $\tau$ is annealed from an initial value of $1$ to a minimum of $0.5$ using an exponential decay rate of $3 \times 10^{-5}$. In the forward pass, we replace the dense $\vec{C}$ with its hard one-hot version while propagating gradients through the relaxed sample using the Straight-Through estimator (Bengio et al., 2013).

### E.3 LOSS FUNCTION

The model is trained by maximizing the Evidence Lower Bound (ELBO), which is bounded by the loss $\mathcal{L}_{\text{ELBO}} = \mathcal{L}_{\text{recon}} + \mathcal{L}_{\text{KL}}$.

RECONSTRUCTION LOSS   Given the decoder's output, the reconstruction loss $\mathcal{L}_{\text{recon}}$ is the binary cross-entropy (BCE) between the input and the output, averaged over the batch:

$$\mathcal{L}_{\text{recon}} = -\frac{1}{B} \sum_{i=1}^{B} \mathbb{E}_{\vec{C} \sim q_{\vec{s}}(\vec{C}|x_i)}[\log f_\psi(x_i|\vec{C})]$$

KL DIVERGENCE   The term $\mathcal{L}_{\text{KL}}$ is the Kullback-Leibler divergence between the approximate posterior $q_{\vec{s}}(\vec{C}|x)$ and a fixed prior $p(\vec{C})$. The prior is a set of $N$ independent, uniform categorical distributions, i.e., $p(\vec{C}_n) = \text{Categorical}([\frac{1}{K}, \ldots, \frac{1}{K}])$ for each latent variable $n$. The KL divergence is calculated analytically, summed over the $N$ variables, and averaged over the batch:

$$\mathcal{L}_{\text{KL}} = \frac{1}{B} \sum_{i=1}^{B} D_{KL}(q_{\vec{s}}(\vec{C}_i|x_i)||p(\vec{C})) = \frac{1}{B} \sum_{i=1}^{B} \sum_{n=1}^{N} D_{KL}(q_{\vec{s}}(\vec{C}_{i,n}|x_i)||p(\vec{C}_n))$$

# F   EXPERIMENTAL DETAILS: REINFORCEMENT LEARNING

This appendix summarizes the experimental setup for the Reinforcement Learning experiments, which assess policy learning in discrete-action Atari environments. The implementation is adapted from the high-quality PPO implementation provided by Huang et al. (2022).

## F.1   ENVIRONMENT

We use the Breakout and Seaquest environments from the Atari Learning Environment (Bellemare et al., 2013), accessed via the Gymnasium library (Towers et al., 2024). The raw game frames undergo a standard preprocessing pipeline using a series of wrappers that, for example:

- Convert images to grayscale and resize them to $84 \times 84$ pixels.
- Stack 4 consecutive frames to capture temporal dynamics
- Clip the rewards to the range $[-1, 1]$ to stabilize training.

This setup is standard for benchmarking performance on Atari games (Mnih et al., 2015).

## F.2   MODEL ARCHITECTURE

The agent employs a shared-parameter actor-critic architecture with a convolutional network backbone:

- Shared Backbone: The network processes the stacked $4 \times 84 \times 84$ input observations, first normalizing pixel values by dividing by 255.0. It then passes through three convolutional layers with ReLU activations. The network architecture is:

    1. 32 filters of size $8 \times 8$ with a stride of 4.
    2. 64 filters of size $4 \times 4$ with a stride of 2.
    3. 64 filters of size $3 \times 3$ with a stride of 1.

    The output is flattened and passed through a fully-connected layer with 512 units (ReLU activated). All layers are initialized using orthogonal initialization.

- Policy and Value Heads: The 512-dimensional latent representation is fed into two separate linear heads:

    - The policy head (actor) outputs a vector of scores, one for each possible action.
    - The value head (critic) outputs a single scalar estimating the state-value.

We test two methods to convert the policy head's scores into action probabilities: the standard *softmax* function and the *catnat* parameterization using the *natural* activation function.

### F.3 PROXIMAL POLICY OPTIMIZATION

The model is trained using the Proximal Policy Optimization (PPO) algorithm (Schulman et al., 2017). PPO is an on-policy algorithm that optimizes a clipped surrogate objective function. The total loss is a combination of the policy loss, the value function loss, and an entropy bonus to encourage exploration:

$$J(\theta) = \hat{\mathbb{E}}_t \left[ L_t^{\text{CLIP}}(\theta) - c_1 L_t^{\text{VF}}(\theta) + c_2\, H[\pi_\theta](s_t) \right],$$

where the clipped surrogate is

$$L_t^{\text{CLIP}}(\theta) = \min\left( r_t(\theta)\, \hat{A}_t,\ \text{clip}\big(r_t(\theta), 1 - \epsilon, 1 + \epsilon\big)\, \hat{A}_t \right),$$

Here, $r_t(\theta) = \frac{\pi_\theta(a_t|s_t)}{\pi_{\theta_{\text{old}}}(a_t|s_t)}$ is the probability ratio, and $\hat{A}_t$ is the advantage estimate. Advantages are calculated using Generalized Advantage Estimation (GAE) (Schulman et al., 2015) with $\gamma = 0.99$ and $\lambda = 0.95$, and are normalized per mini-batch. The value loss is typically

$$L_t^{\text{VF}}(\theta) = \tfrac{1}{2}\big(V_\theta(s_t) - \hat{V}_t\big)^2,$$

and $H[\pi_\theta](s_t)$ denotes the policy entropy. The clipping $\epsilon$ and coefficients $c_1, c_2$ are hyperparameters.

### F.4 HYPERPARAMETER OPTIMIZATION

Due to the high computational cost, we performed a targeted hyperparameter search instead of an exhaustive grid search. For each parameterization method and environment, we ran 160 trials using a Tree-structured Parzen Estimator (TPE) sampler (Bergstra et al., 2011) to find promising hyperparameter configurations. Table 5 summarizes each parameter's type, sampling range, and any non-default scale or step size. The top 10 configurations identified by this search were then trained with 10 different random seeds to ensure more reliable performance statistics.

Table 5: Hyperparameter sweep: types, sampling ranges, and non-default scales/steps.

| Parameter | Type | Range | Scale / Step / Notes |
|---|---|---|---|
| learning_rate | float | $5.0 \times 10^{-5} - 1.0 \times 10^{-2}$ | log scale |
| num_steps | int | $32 - 512$ | step $= 32$ |
| update_epochs | int | $1 - 16$ | step $= 2$ |
| clip_coef | float | $0.01 - 0.90$ | linear sampling |
| ent_coef | float | $0.0 - 1.0$ | linear sampling |
| num_envs | int | $8 - 16$ | step $= 2$ |
| num_minibatches | int | $2 - 16$ | step $= 4$ |
| max_grad_norm | float | $0.1 - 10.0$ | step $= 0.1$ |

### F.5 ADDITIONAL TRAINING PARAMETERS

We trained all models for a total of 8 million timesteps using the Adam optimizer (Kingma & Ba, 2015) with an $\epsilon$ of $10^{-5}$. The learning rate, identified via hyperparameter search, was linearly annealed to zero over the course of training.

## G LARGE LANGUAGE MODEL (LLM) USAGE

In accordance with the guideline requirements[3], we acknowledge that LLMs were employed to refine and rephrase portions of the text. The ideas, their development, the interpretation of the results, and all scientific contributions were carried out solely by the authors.

---

[3]See "The Use of Large Language Models (LLMs)" at https://iclr.cc/Conferences/2026/AuthorGuide

