# OpenReview forum: "Beyond Softmax: A Natural Parameterization for Categorical Random Variables"
_ICLR.cc/2026/Conference — Submitted to ICLR 2026_

### Official Review · Reviewer_qbc1 · 2025-10-31

**Soundness:** 2
**Presentation:** 3
**Contribution:** 2
**Rating:** 2
**Confidence:** 3

**Summary:**

The article studies the problem of learning latent representations of categorical variables. In particular, it examines the role of the softmax function, which maps a score vector to a valid categorical probability vector. The paper shows that this function leads to a dense Fisher Information Matrix (FIM) for a categorical random variable and argues that such dense matrices can introduce geometric distortions that may hinder gradient-based optimization.

To address this, the authors propose a new parameterized function, catnat, for learning categorical probability vectors. This function is designed to yield a diagonal FIM, potentially mitigating the aforementioned issues. Experimental results on three different tasks are presented to illustrate the performance of the proposed method.

**Strengths:**

The strengths of the paper are:
1. The paper examines the widely used softmax function from an Information Geometry perspective, which provides an interesting theoretical viewpoint.

2. A novel parameterized function (catnat) is proposed and analyzed as an alternative to softmax.

3. The experimental results appear to demonstrate improved performance compared to the softmax baseline.

**Weaknesses:**

1. The proposed function introduces additional hyperparameters, which may complicate training and tuning.

2. The advantages and underlying mechanisms of the new function are not clearly articulated.

3. The numerical results are difficult to interpret and do not strongly support the claimed benefits.

**Questions:**

Overall, the paper is well written and offers an interesting new perspective on the softmax function. However, several aspects of the current version need clarification and improvement.

1. Practical Relevance of the Approach:

The paper focuses on the softmax function and proposes a new parameterization for categorical random variables in the context of Information Geometry and Natural Gradient Descent. However, most modern machine learning and AI methods rely on standard gradient descent or stochastic gradient descent (SGD) due to the large scale of neural networks, where computing or inverting the FIM is infeasible. It is therefore unclear whether the proposed analysis and the new function are practically useful.

How does the catnat function perform when used with standard GD or SGD rather than natural gradients?

2. Justification of Claims about FIM and Geometry:

The paper claims that a dense FIM introduces geometric distortions, but it does not adequately explain why this occurs or what these distortions entail. The rationale behind why a diagonal FIM would alleviate this issue is also unclear.

Furthermore, since the activation functions a() and ν() used in catnat can take zero values, the resulting FIM could be singular.
Overall, the motivation for the new function is not fully convincing.

3. Hyperparameters and Computational Cost:

Softmax is simple, widely adopted, and free of hyperparameters. The proposed catnat function introduces parameters A and C, which must be tuned.

The paper should discuss the sensitivity of results to these hyperparameters and the computational cost compared to softmax, especially for large numbers of categories.

The computation of each probability entry in catnat involves multiple products of powers of activation functions, which could be expensive. A complexity analysis would be helpful.

4. Experimental Details and Interpretation:

Several details are missing or unclear in the numerical experiments:

(i) In Table 2, information about the graph sizes and GNN architectures is missing.

(ii) In Table 3, the reported negative log-likelihood values (e.g., 97.8 ± 0.2 vs. 97.5 ± 0.3) are very close, with overlapping error ranges. It is unclear whether such differences are statistically or practically significant.

Similarly, in Table 4, results such as 406 ± 34 vs. 398 ± 25 do not convincingly demonstrate improvement.

Overall, the empirical results do not clearly establish that the proposed function provides a meaningful advantage over softmax, especially given the added complexity and tuning effort.

---

> ### Author Response · Authors · 2025-11-17
>
> Thank you for your review; we address the questions and weaknesses below.
>
> **W1 and Q3**
>
> Please note that the parameters $A$ and $C$ are not additional hyperparameters that require tuning and can be fixed to $C=0$ and $A=2\pi$. Their role is simply to adjust the centering and amplitude of the natural activation function $\nu$, **analogously to how one might shift or scale a sigmoid**. Regarding computational complexity, it remains $O(K)$, identical to the `softmax` function. Further details are provided in "General Comment 2 - Computational Complexity".
>
> **W2 and Q2**
>
> We refer to "General Comment 2 - Why Information Geometry?". If any specific aspect remains unclear after the revision, we welcome further questions
>
>
> **W3**
>
> We agree that in some settings the `softmax` parameterization yields results that are close to, or statistically indistinguishable from, those obtained with the proposed catnat parameterization. Nevertheless, we emphasize the following:
>
> 1) **The `softmax` never outperforms the `catnat` parameterization** in our experiments.
>
> 2) in **$77.5 \\%$ of the reported results, the `catnat` parameterization outperforms the `softmax` by one or more standard deviations**.
>
> We therefore consider the proposed parameterization to be reliably advantageous and a preferable choice when working with categorical random variables. We further view these empirical observations as robust, given the supporting theoretical analysis.
>
> **Q1**
>
> Since the proposed parameterization is grounded in Information Geometry and directly targets issues arising in gradient-based optimization, we found your question particularly relevant. We therefore ran an additional set of experiments replacing the Adam optimizer with SGD to assess whether the results extend to different gradient-based optimization algorithms. Below we report the results for the VAE experiment on the binarized MNIST dataset; the same experiments on the MNIST dataset are currently running.
>
> _Table 3: SGD Binary MNIST_
>
> | N  | Method              | K = 8             | K = 16            | K = 32            |
> |----|---------------------|-------------------|-------------------|-------------------|
> | 10 | softmax             | 86.7 ± 0.7        | 83.4 ± 0.3        | 84.9 ± 2.7        |
> |    | hierarchical σ      | *84.0 ± 0.6*      | **80.1 ± 0.2**   | *81.0 ± 0.5*        |
> |    | hierarchical ν      | **83.8 ± 0.2**    | *81.3 ± 0.4*     | **80.9 ± 0.2**        |
> | 20 | softmax             | 79.6 ± 0.5        | 80.7 ± 0.1        | 86.6 ± 2.9        |
> |    | hierarchical σ      | **78.7 ± 0.5**        | **79.2 ± 0.7**        | *81.5 ± 2.0*        |
> |    | hierarchical ν      | *79.0 ± 0.9*        | **79.2 ± 0.8**        | **80.9 ± 0.9**        |
> | 30 | softmax             | 80.1 ± 0.8        | 81.3 ± 0.5        | 85.1 ± 1.6        |
> |    | hierarchical σ      | **79.5 ± 1.0**        | **79.3 ± 0.6**        | **82.9 ± 1.4**        |
> |    | hierarchical ν      | *79.7 ± 1.2*        | *80.2 ± 1.9*        | *83.5 ± 3.0*        |
>
>
>
> The results indicate that the performance patterns seen with Adam extend to SGD, with the hierarchical parameterizations consistently outperforming the softmax baseline. We have included these results in the paper.
>
>
> **Q4**
>
> (i) We deferred the additional details to Appendix D "Experimental Details: Graph Structure Learning". We used a classic GCN [Kipf 2017] trained on a graph with 4 communities, resulting in $> 500$ edge parameters $\theta_{ij}$ to be learned.
>
> (ii) We refer to **W3**. We remark that no additional computational complexity or tuning is required beyond replacing the parameterization. We will clarify this point more explicitly in the final version of the paper.
>
>
> **References**
>
> Thomas Kipf and Max Welling. Semi-supervised classification with graph convolutional networks. ICLR, 2017.

---

> > ### Author Response · Authors · 2025-11-21
> >
> > We extend our response to **Q1** by adding the results from the corresponding experiment on the MNIST dataset:
> >
> > _Table 4: SGD MNIST_
> >
> > | N  | Method              | K = 8             | K = 16            | K = 32            |
> > |----|---------------------|-------------------|-------------------|-------------------|
> > | 10 | softmax             | 102.3 ± 0.4       | 101.5 ± 0.8       | 102.4 ± 0.5       |
> > |    | catnat $\sigma$      | **100.6 ± 0.3**       | **99.3 ± 0.9**        | **100.1 ± 1.1**       |
> > |    | catnat $\nu$      | **100.6 ± 0.4**       | _99.6 ± 0.6_        | _101.2 ± 1.4_       |
> > | 20 | softmax             | 99.3 ± 0.3        | 100.5 ± 1.1       | 106.1 ± 0.8       |
> > |    | catnat $\sigma$      | **98.6 ± 0.3**        | _99.7 ± 1.4_        | **99.9 ± 0.5**        |
> > |    | catnat $\nu$      | _98.8 ± 0.5_        | **98.8 ± 0.4**        | _100.7 ± 0.4_       |
> > | 30 | softmax             | _99.7 ± 0.4_        | 100.6 ± 0.8       | 110.4 ± 5.7       |
> > |    | catnat $\sigma$      | 99.8 ± 1.3        | **99.4 ± 0.5**        | _104.8 ± 2.7_       |
> > |    | catnat $\nu$      | **99.5 ± 0.2**        | _100.2 ± 0.9_       | **102.0 ± 1.0**       |
> >
> > The results further support the effectiveness of the `catnat` parameterization.

---

### Official Review · Reviewer_4C1Y · 2025-11-02

**Soundness:** 2
**Presentation:** 2
**Contribution:** 2
**Rating:** 2
**Confidence:** 5

**Summary:**

The authors suggest an alternative to the Gumbel-softmax trick for the stochastic gradient estimator of the latent categorical random variables. The proposed catnat is an alternative parameterization of the softmax function, consists of sequential hierarchical binary splits, based on the concept called information geometry and natural gradient. The authors provide theoretical backup for the proposed method, and conduct wide range of experiments.

**Strengths:**

- The algorithmically proposed methodology makes sense.
- The theoretical basis was also appropriately presented.

**Weaknesses:**

- With the current shape of the paper, it is hard to understand how to apply the suggested method practically. Please provide the pseudo-code or algorithm of the proposed method.
- The experiments are wide, but very shallow. It is very hard to tell whether the suggested method is superior to the baselines. Thorough experiments against various stochastic gradient estimators for the categorical random variables should be conducted.

**Questions:**

- How can we implement the proposed method when num_category $\neq 2^n$?
- What is the computational complexity for the suggested method?
- Also, the computation cost should be compared against various baselines. Why are all the baselines missing?
- The following is missing baselines [1,2,3,4], at least for the VAE experiment.
- How does the variance of the stochastic gradient differ from the baselines?

[1] Mnih et al., Neural variational inference and learning in belief networks
[2] Gu et al., Muprop: Unbiased backpropagation for stochastic neural networks
[3] Tucker et al., REBAR: Low-variance, unbiased gradient estimates for discrete latent variable models
[4] Grathwohl et al., Backpropagation through the void: Optimizing control variates for black-box gradient estimation

**Details Of Ethics Concerns:**

.

---

> ### Author Response · Authors · 2025-11-17
>
> Thank you for your time spent writing the review.
>
> We believe there has been a critical misunderstanding of our paper. **We do not propose an alternative to the Gumbel–Softmax trick, nor to any other gradient estimation technique**. Our work studies the problem of training latent categorical random variables from a novel, complementary perspective, and indeed “remains compatible with standard training stabilization techniques” (i.e., different gradient estimation techniques can be used, and are used in the experiments).
>
> Thus, [1,2,3,4] (that we cite in our paper) are not alternatives or baselines to our method; they can be employed together with our parameterization. We hope that this clarification addresses **W2**, **Q3**, **Q4**, and **Q5**. If any aspect remains unclear and further detail would help, we are glad to provide it.
>
> **W1 and Q2**
>
> We refer to the "General Comment 2 - Computational Complexity". If anything remains unclear, we would be happy to elaborate on specific aspects you find missing.
>
>
> **Q1**
>
> When $K \neq 2^n$, the hierarchical binary decision tree (Figure 3) can simply be terminated at level $H-1$ for some nodes. This is equivalent to pairwise merging the final probability masses of some leaves. This corresponds to pairwise merging the final probability masses of some leaves, yielding a valid categorical distribution with exactly $K$ leaves.

---

### Official Review · Reviewer_DU4o · 2025-11-03

**Soundness:** 1
**Presentation:** 3
**Contribution:** 2
**Rating:** 2
**Confidence:** 3

**Summary:**

The paper introduces a parameterisation of Categorical random variables aimed at inducing a diagonal Fisher information matrix, which, the paper argues, is poised to address many of the pitfalls of the typical softmax parameterisation. This works by parameterising the categorical probabilities as the result of a binary branching process (somewhat akin a stick breaking procedure).

The idea is tested on some latent variable modelling problems, though in rather simple settings of graph structure learning, categorical VAE, and an RL setting. Through different performance metrics in these tasks, the paper argues that the proposed approach (catnat) is better suited than softmax.

**Strengths:**

1. I find the parameterisation elegant
2. The paper is mostly rather clear

**Weaknesses:**

1. The paper argues that the proposed technique addresses challenges that countless other papers addressed by various other means, yet none of those, it seems, are relevant for the empirical section. I find this a big omission. Catnat is a novel parameterisation and, as far as the reader can tell, there's so much to be established: stability, computational tradeoffs, performance against other parametrisation aimed at similar issues, performance against reasonable gradient estimators, etc.

2. The paper misses an entire body of literature on sparse parameterisations (I'm thinking of the sparsemax family) for which a lot of work on inducing structure has been published (e.g., entmax, sparsemap, etc.). I'm listing only a fraction of the relevant papers in this line of work below:

- https://proceedings.mlr.press/v48/martins16.html
- http://papers.neurips.cc/paper/6926-a-regularized-framework-for-sparse-and-structured-neural-attention.pdf
- https://proceedings.mlr.press/v80/niculae18a.html

It also misses other sparse parameterisations, for example those based on 'stretch and rectify' principle:

- https://openreview.net/forum?id=H1Y8hhg0b
- https://openreview.net/forum?id=WAid50QschI

Are there worthwhile connections between these techniques and catnat? Do they address similar issues? Are they addressed by fundamentally different things? How do they fare in comparison, etc. The reader cannot tell.

3. The paper claims the proposed parameterisation is poised to address problems of the softmax parameterisation, and that the essence of the solution is that the Fisher information matrix is diagonal, but the paper does not discuss why this is expected to matter for the stability of the techniques used to train LVMs in this paper (ie, policy gradient, PPO, and the gumbel-softmax relaxation). I don't think this can be left for the reader to infer, the paper has to spell out the arguments clearly.

For the reasons above my judgment of soundness is as is.

**Questions:**

Other than the weaknesses above, I have a small question.

Does something like the stick breaking construction, which the paper also fails to acknowledge, induce a Fisher information matrix somewhat like catnat's?

https://openreview.net/pdf?id=S1jmAotxg

---

> ### Author Response · Authors · 2025-11-17
>
> Thank you for the time spent writing the review. It emerged that we did not articulate clearly enough how our paper is positioned within the existing literature, and we are pleased to clarify the novelty of our contribution and to provide additional experiments and references where needed.
>
> **W1 \& W2**
>
> The majority of works that improve optimization for models with latent categorical random variables focus on reducing the variance (or bias) of gradient estimators.
> Our approach is orthogonal to these techniques: it replaces the `softmax`, not the gradient estimator. Thus, comparing the “performance against reasonable gradient estimators” is not directly applicable since these methods can be combined with `catnat`, as also reflected in our experiments, where different estimators were used across different settings.
>
> As noted, there exist alternative parameterizations (e.g., `sparsemax` and `entmax`), but these pursue different goals such as sparsity or interpretability. Our objective is distinct: to improve the optimization dynamics, yielding better performance.
>
> For this reason, **we are not aware of any prior parameterization that aims at the same goal**.
>
> Nevertheless, we agree that comparing against a representative sparse parameterization provides additional context. Therefore, we ran an additional VAE experiment with `sparsemax` and included the results, with `catnat` as reference:
>
> _Table 1: Binary MNIST_
>
> | N  | Method        | K = 8             | K = 16            | K = 32            |
> |----|---------------|-------------------|-------------------|-------------------|
> | 10 | sparsemax     | 87.0 ± 0.4        | 83.4 ± 0.9        | 85.2 ± 1.5        |
> |    | *catnat* ν    | **83.2 ± 0.5**    | **78.7 ± 0.3**    | **77.3 ± 0.4**    |
> | 20 | sparsemax     | 84.3 ± 0.8        | 83.6 ± 0.8        | 85.1 ± 0.6        |
> |    | *catnat* ν    | **77.1 ± 0.4**    | **76.6 ± 0.3**    | **76.8 ± 0.4**    |
> | 30 | sparsemax     | 86.9 ± 0.6        | 87.2 ± 1.0        | 88.7 ± 1.3        |
> |    | *catnat* ν    | **77.9 ± 0.6**    | **77.7 ± 0.5**    | **78.0 ± 0.7**    |
>
> _Table 2: MNIST_
>
> | N  | Method            | K = 8             | K = 16            | K = 32            |
> |----|-------------------|-------------------|-------------------|-------------------|
> | 10 | sparsemax         | 102.5 ± 0.3       | 101.0 ± 0.6       | 101.3 ± 0.8       |
> |      | catnat $\nu$        | **99.8 ± 0.4**    | **97.6 ± 0.2**    | **96.9 ± 0.4**    |
> | 20 | sparsemax         | 102.1 ± 0.7       | 102.1 ± 0.8       | 103.8 ± 0.7       |
> |      | catnat $\nu$       | **97.7 ± 0.2**    | **97.0 ± 0.4**    | **96.9 ± 0.4**    |
> | 30 | sparsemax         | 105.3 ± 0.5       | 103.9 ± 0.5       | 104.9 ± 0.8       |
> |     | catnat $\nu$       | **97.9 ± 0.3**    | **97.6 ± 0.3**    | **97.7 ± 0.8**    |
>
> The results show that **`catnat` still yields the best test performance**, including with the `sparsemax` experiment added. These comparisons have been incorporated into the revised manuscript.
>
> The computational complexity remains $\mathcal{O}(K)$, matching that of the `softmax`; additional details are provided in “General Comment 2 - Computational Complexity”.
>
> **W3**
> Please see “General Comment 2 - Why Information Geometry?”. If there is any remaining point that requires clarification, we are happy to address it.
>
>
> **Q**
> Thank you for sharing that VAE paper. We agree that Dirichlet processes provide a compelling way to model an unbounded latent space in VAEs without requiring a predefined $K$. Nonetheless, we also see sources of instability that can hinder training. In the stick-breaking formulation, the probability $p_k$ is the product of $k$ terms. In contrast, in the catnat parameterization each $p_k$ involves at most $log(K)$ terms, which improves stability for large $K$.
> Despite the conceptual similarities and differences, we do not have results on the FIM of the stick-breaking process. Thank you again for the question.

---

### Author Response · Authors · 2025-11-17
**General Comment 2**

Below we provide a brief comment on the additional experiments conducted during the rebuttal and we address questions raised by more than one reviewer.

**Additional Experiments**
Some remarks from the reviewers prompted us to further evaluate the `catnat` parameterization, and we thank them for this. In response, we incorporated an additional comparison with a representative sparse parameterization (`sparsemax`) and ran a new set of VAE experiments using SGD instead of Adam to assess optimizer dependence. Across these evaluations, **`catnat` remains the strongest-performing parameterization**, confirming the generality of the results.

**Computational Complexity**
For a categorical distribution with $K$ classes, `softmax` requires $\mathcal{O}(K)$ operations.
The naive implementation of **`catnat` has the same linear complexity**: computing the $K$ values $a_i$ and $(1 - a_i)$ is $\mathcal{O}(K)$, and descending the binary tree requires $\mathcal{O}(2K)$ operations. Moreover, `catnat` can be parallelized (as done in our implementation), making it well suited for GPUs. Upon acceptance, we will release the PyTorch implementation, enabling other researchers to easily integrate `catnat` into their pipelines.

**Why Information Geometry?**
A substantial body of literature [Amari 1998a, 1998b, 2000] shows that optimization benefits from working on flat statistical manifolds. When the manifold is highly curved, equal parameter changes can induce disproportionately different changes in the output distribution, making gradient-based optimization inefficient or unstable.

We attempted to convey this intuition in Section 3 (“Information Geometry & Natural Gradient”), but agree that the explanation can be expanded.
Equation (4) explicitly shows the correction needed to account for curvature. When the FIM is dense, the curvature induces $\mathcal{O}(K^2)$ distorsions, harming optimization. A diagonal FIM eliminates the majority of these interactions.

We have revised the introduction on Information Geometry to highlight its relevance in optimization.

**References**

Shun-ichi Amari. Natural gradient works efficiently in learning. Neural computation, 1998a.

Shun-ichi Amari and Scott C Douglas. Why natural gradient? ICASSP, 1998b.

Shun-ichi Amari and Hiroshi Nagaoka. Methods of information geometry. Vol. 191. American Mathematical Soc., 2000.

---

### Author Response · Authors · 2025-11-17
**General Comment 1**

We thank all reviewers for their time and feedback.

It clearly emerged from the review process - and we acknowledge - our inefficacy in communicating both relevance and novelty of what proposed. It follows that our work may have been interpreted as an alternative approach to existing techniques, which is not the case. We intervened on the manuscript to strengthen differences, value and novelty and hope this communication flaw is now resolved. Having this direction in mind we clarify the following:

**1) The goal**

We aim at **improving the effectiveness of the learning process** when training models employing latent categorical random variables. What proposed enables more effective learning optimization procedure - yielding to **better performance**. As such, we don't modify the `softmax` to impose sparsity or interpretability.

**2) The novelty**

Previous works addressing the aforementioned goal primarily focused on developing techniques yielding improved gradient estimates. **Our work does not propose an alternative gradient estimator**. In fact, the `catnat` parameterization is orthogonal to the choice of the gradient estimator and can be combined with any of them. Differently, **our contribution lies in applying Information Geometry to identify a key limitation of the `softmax` parameterization and proposing a new parameterization that is better suited for optimization**. The proved theoretical advantages emerge from improved empirical performance.

**3) The effectiveness and simplicity of the solution**

We emphasize the computational simplicity behind the proposed minimal change. In fact, the new parameterization achieves better results **without a relevant increase of the computational cost**. Furthermore, performance improvements follow by simply replacing the `softmax` function with the `catnat`, **requiring only a single-line code change**.

**4) The broad impact**

The new parameterization can be applied to **any deep learning model involving latent categorical random variables**. Notable examples are VAEs, reinforcement learning and graph neural networks.

---

### Meta-Review · Area_Chair_QR3C · 2025-12-24

**Summary:**

The reviewers have raised the following major concerns:

(1) Related works are not thoroughly discussed, and there are missing benchmarks in the experimental section.

(2) The connection between the Fisher information matrix and the motivation of the work is not sufficiently articulated.

(3) The experiments are shallow.

(4) Missing complexity analysis.

(5) Practical relevance of the approach.

**Reviewer Concerns:**

Several concerns have been partially addressed, including the addition of new experiments and the clarification of computational complexity.

Nevertheless, some major issues remain. In particular, I share reviewer DU4o’s concern that the connection between the diagonal Fisher information matrix and the claimed benefits of replacing softmax is still not clearly articulated, even in the revised manuscript. High-level arguments based primarily on existing literature are not sufficient; more explicit theoretical justification, such as stability or robustness analysis, would significantly strengthen the case for the proposed catnat method.

Moreover, since the paper challenges the widely adopted softmax function, it would benefit from much stronger empirical evidence that clearly demonstrates both the advantages and potential limitations of the proposed approach. In this regard, I agree with the reviewers that the current numerical evaluation is not yet sufficient to persuade practitioners to adopt catnat over softmax. Incorporating additional benchmarks, precise runtime comparisons, and more carefully designed experiments would make the empirical validation more compelling.

**Reviewer Scores:**

While there is no direct indication that the reviewers will revise their scores, the work does present an interesting alternative to softmax, and several concerns have been partially addressed. As such, there remains some possibility that one or more reviewers may adjust their evaluations upward. That said, the paper would benefit from a more comprehensive theoretical analysis and more detailed numerical studies to further strengthen its claims. In its current form, however, the work is not yet ready for acceptance.

---

### Decision · Program_Chairs · 2026-01-26

Reject